# Palladium-catalysed synthesis of triaryl(heteroaryl)methanes

Shuguang Zhang[1], Byeong-Seon Kim[1], Chen Wu[1], Jianyou Mao[2] & Patrick J. Walsh[1,2]

Tetraarylmethane derivatives are desirable for a variety of applications, but difficult to access with modern C–C bond-forming reactions. Here we report a straightforward method for palladium-catalysed arylation of aryl(heteroaryl)methanes and diaryl(heteroaryl)methanes with aryl chlorides. This reaction enables introduction of various aryl groups to construct triaryl(heteroaryl)methanes via a C–H functionalization in good to excellent yield, and represents the first step towards a general transition metal catalysed synthesis of tetraarylmethanes.

[1] Roy and Diana Vagelos Laboratories, Department of Chemistry, University of Pennsylvania, 231 South 34th Street, Philadelphia, Pennsylvania 19104-6323, USA. [2] Institute of Advanced Synthesis, School of Chemistry and Molecular Engineering, Jiangsu National Synergetic Innovation Center for Advanced Materials, Nanjing Tech University, 30 South Puzhu Road, Nanjing 211816, China. Correspondence and requests for materials should be addressed to P.J.W. (email: pwalsh@sas.upenn.edu) or to J.M. (email: ias_jymao@njtech.edu.cn).

etraarylmethanes and related derivatives are important building blocks, with uses ranging from molecular devices[1–4] to porous organic frameworks[5–7] and applications from protein translocation detection[8] to drug delivery[9]. Furthermore, a recent study of 9,000 bioactive compounds by AstraZeneca to evaluate molecular space concluded that most biologically active compounds are linear or disk shaped, and very few are sphere-like molecules. They concluded that chemists should '… expand the current arsenal of tools to access less populated space …' and '… this may prove advantageous as the pharmaceutical industry ventures into new disease areas and new target classes which require different molecular shapes to bind and achieve the desired effect'[10]. Tetraarylmethanes are members of sphere-like molecules that have not been widely explored due to synthetic difficulties.

The classic methods to synthesize tetraarylmethanes are based on Friedel-Crafts arylations (Fig. 1a)[11–18] and nucleophilic addition of alkyllithium or Grignard reagents to benzophenone derivatives (Fig. 1b)[19–21]. Each of these approaches has well-known limitations. Friedel-Crafts reactions require arenes with electron-donating groups, often afford mixtures of regioisomers, and are not suitable when *meta*-substituted products are required. Aryl organometallic reagents are typically highly reactive, sensitive to traces of air and moisture, and exhibit limited functional group tolerance.

Transition metal-catalysed cross-coupling reactions have emerged as an excellent method to construct C–C bonds. For example, they have been used with great success in the coupling of aryl halides to diarylmethanes to afford triarylmethanes[22–25] (Fig. 1c)[26]. The application of transition metal catalysts to the synthesis of tetraarylmethanes, however, has proven quite challenging. This may be due to difficulties associated with the transmetallation of the bulky triarylmethyl organometallic species. Thus, there is tremendous unrealized potential in the transition metal catalysed construction of tetraarylmethane derivatives.

We are only aware of a handful of metal catalysed syntheses of tetraarylmethane derivatives. In their synthesis of triarylmethanes, Yorimitsu, Oshima and co-workers noted formation of 8% yield of a tetraarylmethane (Fig. 1c). Despite the intriguing nature of this byproduct, subsequent reports have not been forthcoming. Ghosh and co-workers reported the nickel-catalysed reaction of carbon tetrachloride and PhMgCl to form a 3:2 ratio of triphenylmethyl chloride to tetraphenylmethane (Fig. 1d)[27]. These products were not isolated. Recently, the palladium-catalysed arylation of fluorenes by the teams of Wu and Song, Xie, and Huang was reported to give diarylfluorenes (Fig. 1e)[28,29]. Fluorene is more acidic and less sterically demanding than other diarylmethane derivatives. In significant work, the team of Nambo and Crudden outlined the generation of triarylacetonitriles, $Ar_3C–CN$, then assemble a 4th heteroaryl group by building on the nitrile (Fig. 1f)[30].

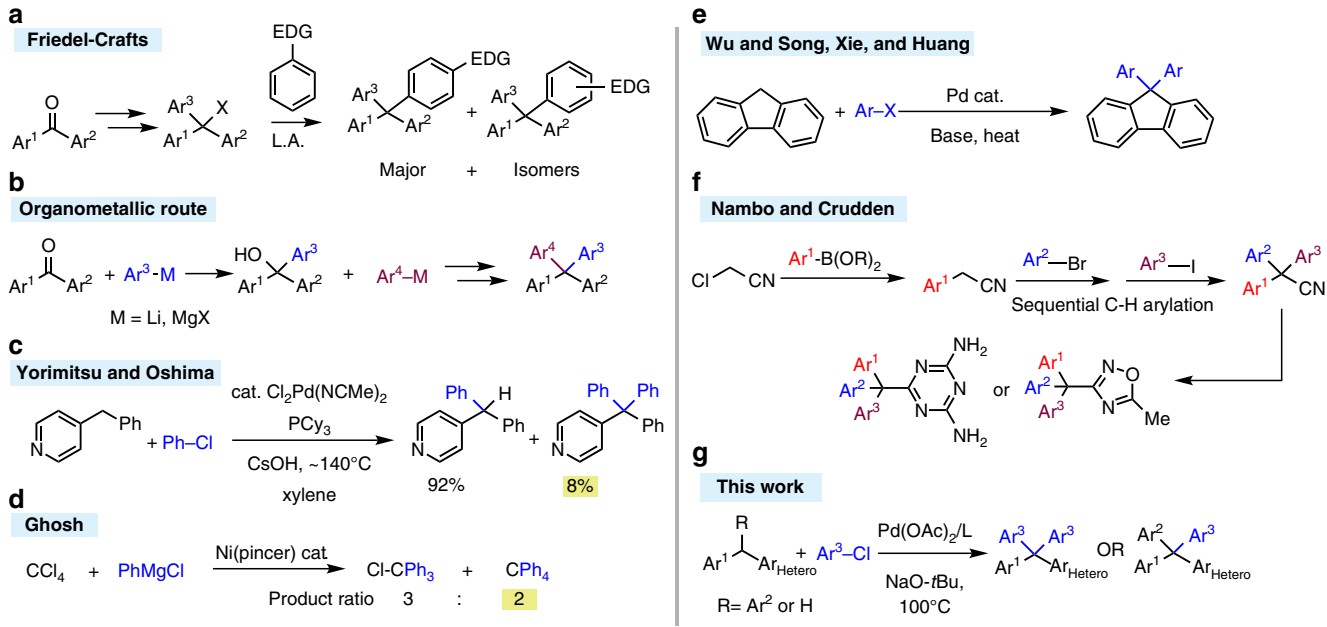

**Figure 1 | Synthesis of tetraarylmethanes. (a,b)** Classic approaches to tetraarylmethanes include the Friedel-Crafts electrophilic aromatic substitution and organometallic additions to triarylmethyl cation precursors. (**c**) Formation of 8% tetraarylmethane byproduct in the synthesis of triarylmethanes. (**d**) Arylation of carbon tetrachloride resulted in up to 39% conversion to tetraphenylmethane. (**e**) Arylation of fluorene. (**f**) Sequential arylation/cycloaddition. (**g**) This work: arylation of 4-benzyl pyridine and (heteroaryl)diphenylmethanes to yield tetraarylmethane derivatives.

**Figure 2 | Ligands screening.** Preliminary reaction screen of 48 ligands.

**Table 1 | Optimization of diarylation of 1a with 2b or 2d\*.**

| entry | L | Pd/L (mol%) | Assay yield (%)[†] |
|---|---|---|---|
| 1 | L1 | 10/20 | 68[‡] |
| 2 | L2 | 10/20 | 77[‡] |
| 3 | L2 | 10/20 | 76[§] |
| 4 | L1 | 10/20 | 66 |
| 5 | L2 | 10/20 | 74 |
| 6 | L1 | 5/10 | 67 |
| 7 | L1 | 2.5/5 | 35 |

*Reactions performed using 1 equiv. of 1a, 4 equiv. of 4-chlorotoluene 2d and 4 equiv. of KOt-Bu on a 0.25 mmol scale.
[†]Yields determined by [1]H NMR analysis of crude mixtures with $CH_2Br_2$ as internal standard.
[‡]Reaction performed using 4-bromotoluene 2b.
[§]Reaction performed using 4-Iodotoluene 2c, isolated yield.

**Table 2 | Optimization of diarylation of 4-benzylpyridine 1a with 4-chlorotoluene 2d\*.**

| entry | Conc (M) | solvent | base | 1a:2d:3 | Assay yield (%)[†] |
|---|---|---|---|---|---|
| 1 | 0.1 | toluene | LiOt-Bu | 1:4:4 | 10 |
| 2 | 0.1 | toluene | NaOt-Bu | 1:4:4 | 83 |
| 3 | 0.1 | toluene | $LiN(SiMe_3)_2$ | 1:4:4 | 17 |
| 4 | 0.1 | toluene | $NaN(SiMe_3)_2$ | 1:4:4 | 58 |
| 5 | 0.1 | toluene | $KN(SiMe_3)_2$ | 1:4:4 | 19 |
| 6 | 0.05 | toluene | NaOt-Bu | 1:4:4 | 12 |
| 7 | 0.2 | toluene | NaOt-Bu | 1:4:4 | 86 |
| 8 | 0.4 | toluene | NaOt-Bu | 1:4:4 | 84 |
| 9 | 0.2 | THF | NaOt-Bu | 1:4:4 | 90 |
| 10 | 0.2 | DME | NaOt-Bu | 1:4:4 | 88 |
| 11 | 0.2 | CPME | NaOt-Bu | 1:4:4 | 81 |
| 12 | 0.2 | 1,4-dioxane | NaOt-Bu | 1:4:4 | 82 |
| 13 | 0.2 | THF | NaOt-Bu | 1:4:4 | 37[‡] |
| 14 | 0.2 | THF | NaOt-Bu | 1:2:2 | 86 |
| 15 | 0.2 | THF | NaOt-Bu | 1:3:3 | 96(92[§]) |
| 16 | 0.2 | THF | NaOt-Bu | 1:3:4 | 92 |
| 17 | 0.2 | THF | NaOt-Bu | 1:4:2 | 85 |
| 18 | 0.2 | THF | NaOt-Bu | 1:4:3 | 93 |

*Reactions performed using Pd(OAc)2 (5 mol%), PCy3 (10 mol%), 1.0 equiv. of 1a and 4 equiv. of 2d on a 0.25 mmol scale.
[†]Yields determined by [1]H NMR analysis of crude mixtures with $CH_2Br_2$ as internal standard.
[‡]The reaction temperature was 80 °C.
[§]Isolated yield.

Herein, we present a novel palladium-catalysed method for the synthesis of heteroaryl-substituted tetraarylmethane derivatives (Fig. 1g). This effort represents the first high-yielding transition metal catalysed preparation of tetraarylmethane derivatives. It enables the synthesis of a broad range of triaryl(heteroaryl)-methanes, including those with four different aryl groups attached to the central carbon atom.

## Results

**Preliminary catalyst screening**. Given that the only example of arylation of a triarylmethane derivative we are aware of is

Yorimitsu, Oshima and co-workers' 8% yield of triphenyl(4-pyridyl)methane (Fig. 1c), we chose this starting point for our reaction design. Instead of CsOH, we employed KOt-Bu, because of the greater solubility of this base. We reasoned that this would also allow us to lower the reaction temperature, and thus use toluene as the solvent. We initiated a search for a suitable catalyst for bis-arylation of 4-benzylpyridine (1a) with bromobenzene (2a) by examining phosphine ligands on microscale using 10 μmol of 1a (see Supplementary Table 1). Thus, 48 electronically diverse mono- (20 mol%) and bidentate phosphine ligands (10 mol%) were tested using Pd(OAc)2 (10 mol%) and 4 equiv KOt-Bu. Reactions were conducted at 110 °C for 12 h, cooled, and

**Table 3 | Scope of aryl chlorides 2 in benzylic C–H bis-arylation of aryl(4-pyridyl)methanes 1\*.**

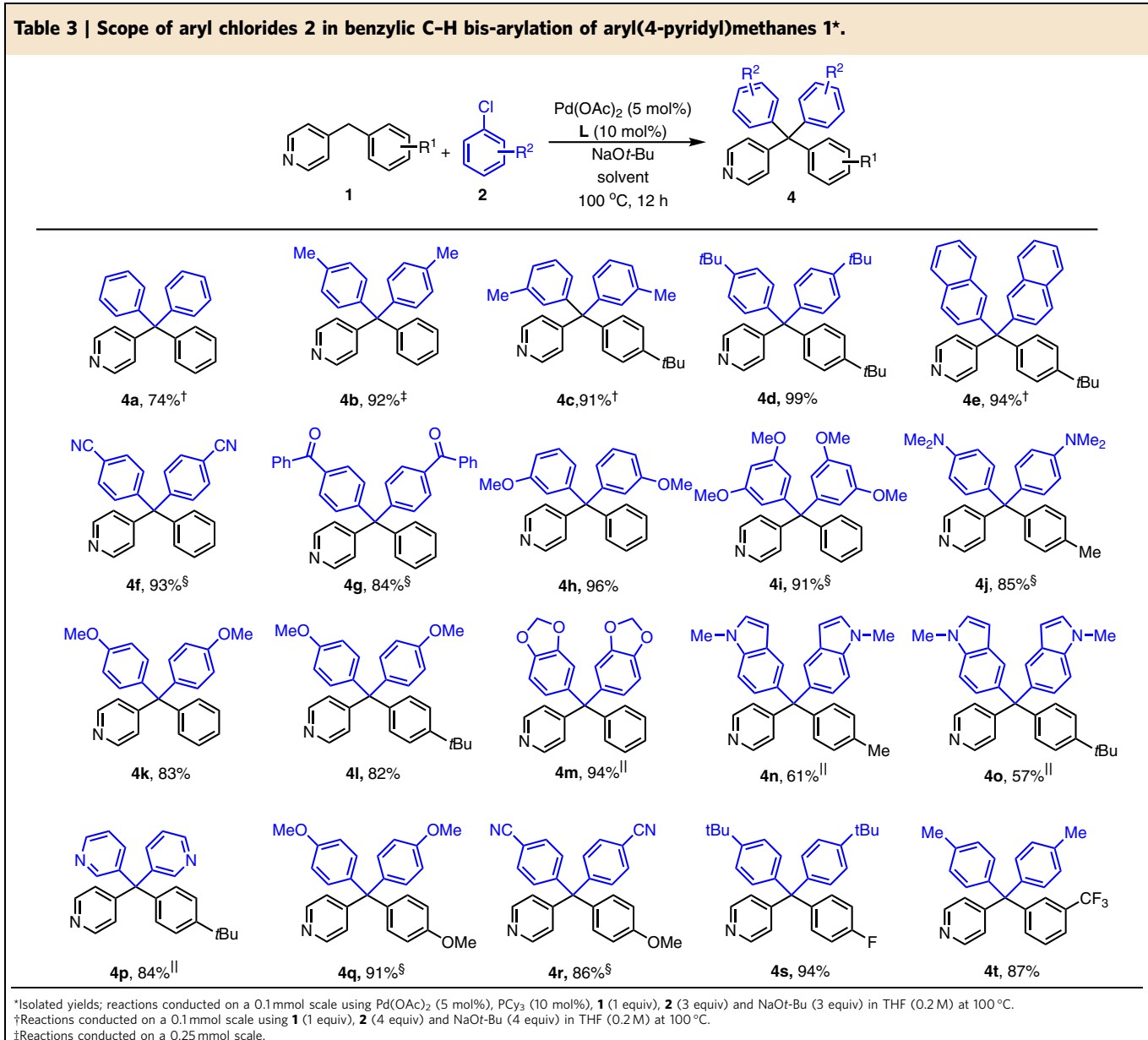

*Isolated yields; reactions conducted on a 0.1 mmol scale using Pd(OAc)$_2$ (5 mol%), PCy$_3$ (10 mol%), **1** (1 equiv), **2** (3 equiv) and NaO$t$-Bu (3 equiv) in THF (0.2 M) at 100 °C.
†Reactions conducted on a 0.1 mmol scale using **1** (1 equiv), **2** (4 equiv) and NaO$t$-Bu (4 equiv) in THF (0.2 M) at 100 °C.
‡Reactions conducted on a 0.25 mmol scale.
§Reactions conducted on a 0.1 mmol scale using Pd(OAc)$_2$ (5 mol%), cataCXium A (10 mol%), **1** (1 equiv), **2** (4 equiv) and NaO$t$-Bu (4 equiv) in 1,4-dioxane (0.2 M) at 100 °C.
‖Reactions conducted on a 0.1 mmol scale using Pd(OAc)$_2$ (5 mol%), cataCXium A (10 mol%), **1** (1 equiv), **2** (4 equiv) and NaO$t$-Bu (4 equiv) in 1,4-dioxane (0.1 M) at 100 °C.

quenched with water. The most promising microscale results, based on product to internal standard (4,4′-di-*tert*-butylbiphenyl) ratios, were with PCy$_3$ (**L1**) and cataCXium A (**L2**, Fig. 2).

Repeating the bis-arylation reaction in Fig. 2 using 4-benzylpyridine **1a** and 4-bromotoluene **2b** with PCy$_3$ (**L1**) and cataCXium A (**L2**) on laboratory scale (0.25 mmol) led to the desired product in 68 (PCy$_3$) and 77% AY (cataCXium A) after 12 h (AY = assay yield, determined by $^1$HNMR analysis, see the Methods section for details) (Table 1, entries 1–2). With cataCXium A as the ligand, 4-iodotoluene **2c** reacted with 4-benzylpyridine **1a** to afford tetraarylmethane product in 76% isolated yield (entry 3). Considering that aryl chlorides are more abundant and less expensive than aryl bromides, we examined aryl chlorides. In the event, the reaction of 4-benzylpyridine **1a** with 4-chlorotoluene **2d** using the same conditions as above was conducted. Interestingly, the diarylated products were obtained in nearly identical assay yields with the aryl chlorides (66% PCy$_3$ and 74% cataCXium A, entries 4–5) and aryl bromides. Although cataCXium A exhibited better AY than

PCy$_3$, we chose to first optimize the reaction with the more economical and readily available PCy$_3$. As outlined below, more challenging substrates gave better results with cataCXium A. To begin with, the catalyst loading was examined. Changing the loading from 10 to 5 mol% resulted in no change in AY (entry 6). Further reducing the amount of catalyst to 2.5 mol%, however, resulted in a drop to 35% (entry 7).

We next focused on the bases (LiO$t$-Bu, NaO$t$-Bu, LiN(SiMe$_3$)$_2$, NaN(SiMe$_3$)$_2$ and KN(SiMe$_3$)$_2$) and reaction concentration (Table 2, entries 1–8). LiO$t$-Bu and LiN(SiMe$_3$)$_2$ gave low yields (10–17%), probably because their stable aggregates result in less reactive bases (entries 1 and 3)[31]. NaN(SiMe$_3$)$_2$, and KN(SiMe$_3$)$_2$, are stronger bases, but resulted in some decomposition and lower yields (entries 4 and 5). The best combination from this screen was 5 mol% Pd(OAc)$_2$, 10 mol% PCy$_3$ and NaO$t$-Bu in toluene (0.2 M) for 12 h at 100 °C (entry 2). Solvents play an important role in deprotonative cross-coupling reactions. Therefore, four additional solvents (THF, DME, CPME and 1,4-dioxane) were examined at 100 °C.

**Table 4 | Scope of aryl chlorides 2 in benzylic C–H arylation of diaryl(4-pyridyl)methanes 5*.**

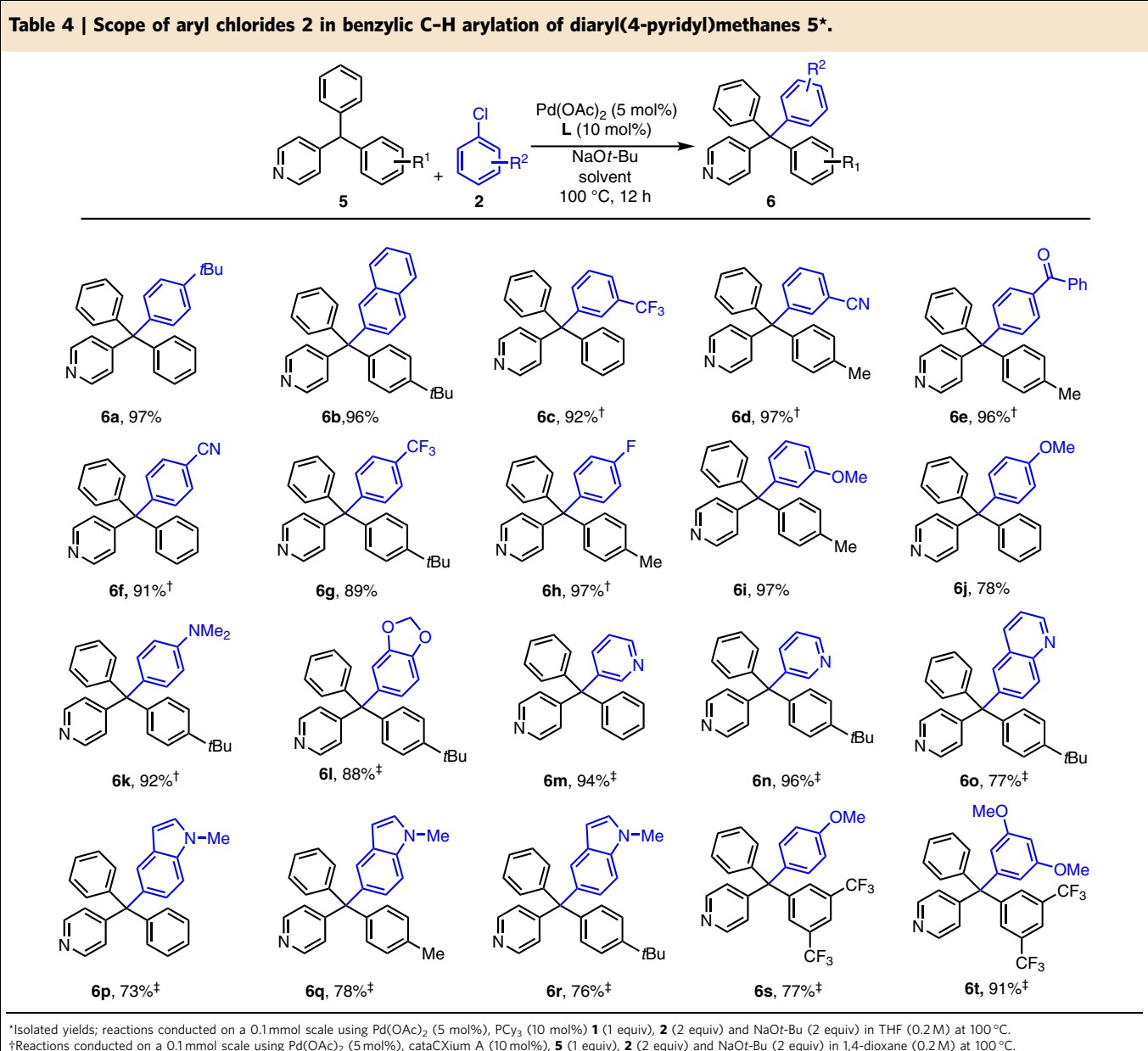

*Isolated yields; reactions conducted on a 0.1 mmol scale using Pd(OAc)$_2$ (5 mol%), PCy$_3$ (10 mol%) **1** (1 equiv), **2** (2 equiv) and NaO*t*-Bu (2 equiv) in THF (0.2 M) at 100 °C.
†Reactions conducted on a 0.1 mmol scale using Pd(OAc)$_2$ (5 mol%), cataCXium A (10 mol%), **5** (1 equiv), **2** (2 equiv) and NaO*t*-Bu (2 equiv) in 1,4-dioxane (0.2 M) at 100 °C.
‡Reactions conducted on a 0.1 mmol scale using Pd(OAc)$_2$ (5 mol%), cataCXium A (10 mol%), **5** (1 equiv), **2** (2 equiv) and NaO*t*-Bu (2 equiv) in 1,4-dioxane (0.1 M) at 100 °C.

Among these, THF led to 90% AY of the diarylation products (entries 9–12). Reducing the reaction temperature to 80 °C resulted in a decrease in the AY to 37% (entry 13). The 4-benzylpyridine: 4-chlorotoluene: base ratio was next explored (entries 14–18). The use of a 1:3:3 ratio rendered 96% AY and 92% isolated yield (entry 15).

**Diarylation of aryl(4-pyridyl)methanes**. Under the optimized reaction conditions (Table 2, entry 15), the deprotonative cross-coupling process (DCCP) with various aryl(4-pyridyl)methanes and aryl chlorides generally gave products in good to excellent yields (Table 3). Coupling of 4-benzylpyridine and alkyl substituted derivatives with chlorobenzene or analogues possessing alkyl substituents in the meta or para positions resulted in products **4a–4e** in 74–99% yield. Coupling with sensitive substrates, such as 4-chlorobenzonitrile and 4-chlorobenzophenone, required cata-CXium A, furnishing 93 and 84% yield of **4f** and **4g**, respectively. In the absence of the palladium catalyst, 4-chlorobenzophenone gave 33% AY of the triarylmethane and *no tetraarylmethane product*.

Using cataCXium A, reaction with 3-chloroanisole and 1-chloro-3,5-dimethoxybenzene provided the bis-arylated products **4h** and **4i** in 96 and 91% yield, respectively. Good yields were obtained using electron rich 4-chloro anisole (82–83%, **4k–l**). Using 4-chloro-*N,N*-dimethylaniline under the standard conditions provided **4j** in 85% yield (cataCXium A as the ligand). Heterocyclic compounds are present in many pharmaceuticals. The heterocyclic catechol derivative **4m** formed in 94% yield (cataCXium A). With cataCXium A, 3-pyridylchloride and 5-chloro-1-methyl-1*H*-indole underwent reaction to give products **4n–4p** in 57–84% yield. 4-Benzylpyridines possessing benzyl groups with 4-OMe, 4-F and 3-CF$_3$ furnished products in 86–94% yield (**4q–4t**) (**4q** and **4r** employed cataCXium A). It is noteworthy that many of these yields are high, despite undergoing two coupling events.

**Monoarylation of diaryl(heteroaryl)methanes**. The ability to synthesize tetraarylmethanes with four different aryl groups would provide greater synthetic flexibility. Based on the successful bis-arylation of 4-benzyl pyridines above, we explored the

monoarylation of diaryl(4-pyridyl)methanes **5** (Table 4). Triarylmethanes are readily prepared by arylation of diphenylmethane derivatives using our prior approach (see Supplementary Methods)[32].

Employing diaryl(4-pyridyl)methanes with two phenyl groups or one phenyl and one alkyl substituted aryl, aryl chlorides with alkyl or neutral groups furnished **6a** and **6b** in 97 and 96% yield, respectively. Aryl chlorides with 3-CN, 3-CF₃, 4-COPh, 4-CN, 4-CF₃, 4-F and 3-OMe reacted with 89–97% yield (**6c**–**6i**). Aryl chlorides bearing electron-donating groups 4-OMe and 4-NMe₂, as well as catechol, underwent coupling in 78–92% yield (**6j**–**6l**). Heterocyclic aryl chlorides, including 3-pyridyl, 5-(*N*-methyl indole), and 6-quinolyl participated in the coupling with cataCXium A in 73–96% yield (**6m**–**6r**). Diaryl(4-pyridyl)-methanes with a phenyl and 3,5-bis-CF₃ aryl groups substituted aryl also participated in DCCP to produce the desired products **6s** and **6t** in 77 and 91% yield, respectively. In addition to the high yields generally observed in the synthesis of tetraaryl-methane derivatives, it is noteworthy that many of the products contain chiral quaternary centres.

Due to the significance of heterocycles in drug discovery and in material science, we chose diphenyl(2-benzothiazolyl)methane (**8a**), diphenyl(2-benzoxazolyl)methane (**8b**) prepared from 2-methylthiazole (**7a**), 2-methylbenzoxazole (**7b**) using the literature approach[24] (see General Methods A and Supplementary Methods) and diphenyl(2-pyridyl)methane (**8c**) to expand the scope of our tetraarylmethane synthesis (Table 5). Heteroaryl chlorides such as 3-chloropyridine and 6-chloroquinoline underwent coupling with diphenyl(heteroaryl)methanes (**8**), to furnish tetraarylmethane derivatives (**9a**, **9b**, **9e** and **9f**) in high isolated yields (86–94%). In addition, 5-chlorobenzothiophene and 1-(4-chlorophenyl)-1*H*-pyrrole proved to be suitable aryl chlorides, furnishing products **9c** and **9d** in 61 and 63% yield, respectively. Diphenyl(2-pyridyl)methane coupled with 4-chlorotoluene to give **9g** in 93% yield. With this catalyst system, less acidic diphenyl(3-pyridyl)methane did not afford the desired product under the optimized reaction conditions.

To demonstrate the potential utility of our method, we performed a gram scale reaction of 4-benzylpyridine **1a** with 4-chlorobenzophenone **2j**. The desired product **4g** was obtained in 79% yield (1.88 g, Fig. 3), demonstrating the reaction is scalable (see Supplementary Methods).

Kato and co-workers reported a liquid-crystalline bowl-shaped molecule that form columnar and micellar cubic structures, using triary(4-pyridyl)methane moieties as building blocks[18]. In order to apply our method, we synthesized the same compound (Fig. 4) from the reaction product **4q** (Table 3). It is noteworthy that Kato synthesized triary(4-pyridyl)methane was based

**Table 5 | Scope of aryl chlorides 2 in benzylic C–H arylation of diphenyl(heteroaryl)methanes 8*.**

*Isolated yields; reactions conducted on a 0.2 mmol scale using Pd(OAc)₂ (5 mol %), cataCXium A (10 mol %), **8** (1 equiv), **2** (2 equiv) and NaO*t*-Bu (2 equiv) in 1,4-dioxane (0.1 M) at 100 °C.

**Figure 3 | Reaction on gram scale.** Diarylation of 4-benzylpyridine with 4-chlorobenzophenone on gram scale.

**Figure 4 | Transformation of reaction product.** Synthesis of liquid crystal former **11**.

on Friedel-Crafts arylations from diaryl(4-pyridyl)methane with 46% yield, which is less than our method (91% yield, see Supplementary Methods).

## Discussion

Our research team has been interested in the catalytic functionalization of weakly acidic sp$^3$-hybridized C–H bonds through a DCCP. These reactions involve a weakly acidic C–H of the substrate (pronucleophile) that is reversibly deprotonated under the reaction conditions. Subsequently, the nucleophile undergoes catalyst promoted arylation[33–36] or vinylation[37]. This method has been particularly successful with the generation of triarylmethanes[38–40]. Tetraarylmethane derivatives are challenging to efficiently prepare by both classical and state-of-the-art methods. The breadth of their applications has outpaced chemists' ability to prepare them in a concise fashion. Cross-coupling methods represent an appealing approach to tetraarylmethane derivatives, but to date successful reports of such processes are still lacking. Outlined herein is a palladium-catalysed DCCP for direct arylation of aryl(heteroaryl)methanes and diaryl(heteroaryl)methanes with aryl chlorides to provide triaryl(heteroaryl)methanes. Unlike traditional cross-coupling procedures, which employ prefunctionalized coupling partners, our approach relies on reversible deprotonation of the diarylmethane derivatives under the conditions used for the catalytic C–C bond forming reaction. Under our reaction conditions, a variety of triaryl(heteroaryl)methanes were prepared in good to excellent yields. This communication represents the first steps towards our goal of developing metal catalysed approaches for the construction of a wide range of tetraarylmethanes.

## Methods

**General procedure A.** An oven-dried 8.0 ml reaction vial equipped with a stir bar was charged with 2-methylthiazole (**7a**, 0.50 mmol, 1.0 equiv) or 2-methylbenzoxazole (**7b**, 0.50 mmol, 1.0 equiv) and chlorobenzene (**2e**, 3.0 equiv) in a glove box under a nitrogen atmosphere at room temperature. A stock solution containing Pd(OAc)$_2$ (5.6 mg, 0.025 mmol, 5 mol%) and PCy$_3$ (14.0 mg, 0.05 mmol, 10 mol%) in dry $o$-xylene (2.5 ml). Then, NaO$t$-Bu (3.0 equiv) was added to the reaction mixture. The vial was capped, removed from the glove box, and heated to 130 °C for 12 h with stirring. The reaction mixture was quenched with 0.5 ml of H$_2$O, diluted with 10 ml of ethyl acetate, and filtered over a pad of MgSO$_4$ and silica. The pad was rinsed with ethyl acetate (10 × 2 ml) and the combined solutions were concentrated *in vacuo*. The crude material was loaded onto a deactivated silica gel column and purified by flash chromatography to afford the desired products **8a** and **8b** in 81 and 49% yield, respectively.

**General procedure B.** An oven-dried 8 ml reaction vial equipped with a stir bar was charged with aryl(4-pyridyl)methanes (**1**, 0.10 mmol, 1.0 equiv) or diaryl(heteroaryl)methanes (**5**, 0.10 mmol, or **8**, 0.20 mmol, 1.0 equiv) and aryl chlorides (**2**, 2.0–4.0 equiv) in a glove box under a nitrogen atmosphere at room temperature. A stock solution containing Pd(OAc)$_2$ (1.1 mg, 0.005 mmol, 5 mol%) and PCy$_3$ (2.8 mg, 0.01 mmol, 10 mol%) in dry THF or cataCXium A (3.6 mg, 0.01 mmol, 10 mol%) in dry 1,4-dioxane was taken up by syringe and added to the reaction vial under nitrogen. Then, NaO$t$-Bu (2.0–4.0 equiv) was added to the reaction mixture. The vial was capped, removed from the glove box, and heated to 100 °C for 12 h with stirring. The reaction mixture was quenched with three drops of H$_2$O, diluted with 3 ml of ethyl acetate, and filtered over a pad of MgSO$_4$ and silica. The pad was rinsed with ethyl acetate (3 × 2 ml) and the combined solutions

were concentrated *in vacuo*. The crude material was loaded onto a deactivated silica gel column and purified by flash chromatography to afford the desired products.

**Data availability.** The authors declare that the data supporting the findings of this study are available within the article and its Supplementary Information files. For the experimental procedures and spectroscopic and physical data of compounds, see Supplementary Methods. For $^1$H and $^{13}$C{$^1$H} NMR spectra of compounds, see Supplementary Figs 1–102.

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

## Acknowledgements

We gratefully acknowledge the financial support from Nanjing Tech University and SICAM Fellowship from Jiangsu National Synergetic Innovation Center for Advanced Materials. We thank the National Science Foundation (CHE-1464744) and National Institutes of Health (NIGMS 104349) for financial support.

## Author contributions

S.Z. performed the experiments, B.-S.K. and J.M. performed the initial reaction development. C.W. performed screening experiments. P.J.W. devised the project and provided overall supervision. All authors contributed to writing the manuscript.

## Additional information

**Competing financial interests:** The authors declare no competing financial interests.

**Publisher's note**: 

