## [Peer review file · Nature Communications]

Reviewers' comments:

Reviewer #1 (Remarks to the Author):

The authors report palladium-catalyzed mono- or bis-arylation of aryl(4-pyridyl)methanes or diaryl(4-pyridyl)methanes affording tetraarylmethanes. Tetraarylmethanes are important synthetic building blocks in modern organic chemistry, and a number of approaches to this important class of compounds have been developed. Catalytic arylation of the sp³-hybridized C-H bonds of diarylmethanes or triarylmethanes would be potentially the most straightforward process. In this manuscript, the authors achieved the cross-coupling reactions of aryl(4-pyridyl)methanes or diaryl(4-pyridyl)methanes with aryl chlorides under simple palladium/phosphine catalysis. The coupling reactions proceeded smoothly with a variety of aryl chlorides having an electron-donating or -withdrawing group. This is an interesting piece of synthetic organic chemistry. However, I hesitate to recommend Publication in Nature Commun. because of the following reasons.

- 1) The nucleophilic coupling partner must have a 4-pyridyl moiety, which is a significant limitation that diminishes the value of this transformation.
- 2) Although tetraarylmethanes are important in organic chemistry, the impact of these molecules is rather limited. It is necessary to add one or two examples that justify that this transformation is really useful.

Additional minor comments:

- 1) At least, the outcomes of the coupling reactions with 2- and 3-pyridyl substituted diarylmethanes or triarylmethanes should be described. Discussion about the effect of the 4-pyridyl moiety is necessary.
- 2) How about aryl bromides, iodides, or sulfonates under the optimal reaction conditions?
- 3) Page 2: 2aa and 2ba should be 2a and 2b.

Reviewer #2 (Remarks to the Author):

In this manuscript, Zhang, Kim, Wu, Mao, and Walsh describe a powerful palladium-catalyzed deprotonative cross-coupling process to deliver tetraaryl alkanes. This reaction solves a current challenge in the preparation of tetraaryl alkanes; it enables synthesis of these molecules with 4 different aryl substituents. It also does not require prefunctionalization of the benzylic nucleophile; deprotonation generates the nucleophile in situ. The yields are excellent, as is the incorporation of heterocycles. The functional group tolerance is also fairly broad. The Supporting Information is thorough and well prepared. I anticipate that the synthetic organic and catalysis communities, as well as those using tetraaryl alkanes in materials and biomedical applications, will be very excited by this new method. I recommend publication after the following very minor revisions have been addressed.

- Given the discussion of solvent in the optimization, please give the solvent in Fig 1C for comparison.
- It is a distinct bond disconnection, but Crudden's method to make triaryl acetonitriles may be worth citing. It uses a transition metal-catalyzed approach to install the aryl substituents, and the nitrile can be converted to a nitrogen heterocycle, ultimately providing access to tetraarylmethanes with 4 distinct substituents.
- In the first paragraph of the Results, it would be helpful to remind readers that the pyridyl group is necessary to attain the requisite acidity of the benzylic protons. Of course, it is also a desirable heteroaromatic from a synthetic perspective.
- In the Preliminary Catalyst Screen (line 77), the internal standard is CH₂Br₂, but the Supporting Information says that 4,4'-di-tert-butylbiphenyl was used.
- Table 2 has two footnote c's.
- How are triaryl starting materials (5) prepared? A brief comment would be helpful to show that these are readily accessible.

Reviewer #3 (Remarks to the Author):

The manuscript " Palladium-Catalyzed Synthesis of Triaryl(4-pyridyl)methanes" by Walsh et al. describes the application of their deprotonative coupling methodology to the preparation of pyridyl-substituted triaryl methanes. The authors are certainly correct in their assertion that there are few if any metal catalyzed methods for the preparation of tetraaryl methanes. There is one that is missing from their list and that is the work of Nambo (ref 25) since high yielding routes to tetraaryl methanes are included in this publication, although limited to certain types of heterocycles as one of the rings that can be formed by a cycloaddition. The Walsh report is certainly the most general of the few methods available and one of the only general transition metal catalyzed routes.

The main criticism I have of the paper is the lack of any rationale for the inclusion of the pyridyl substituent. Is this necessary? There are no examples without pyridine in the starting material, leading to the question of whether this is needed for the reaction or not. In addition, the diversity of scope of the aryl groups incorporated is quite narrow, no furans or thiophenes are included and no pyrroles or N-heterocycles other than pyridine or indoles are included. In my opinion, these are the major negatives of the paper. The development of a route to tetraaryl methanes is highly impressive, but work should be done to really show the scope of the method since it is competing against existing methods with substrate limitations (organolithium and Friedel Crafts).

On this same vein, did the authors try a sequential, one pot bis arylation of the substrate? This would dramatically improve the method. There is no doubt that routes to unsymmetrical tetraaryl methanes are infinitely more valuable than routes to symmetrical versions. One could envision using Walsh chemistry to affect back to back additions of different aryl halides in one pot, this should be attempted and the results reported on.

In this reviewer's opinion, the paper would read more cleanly if results and discussion were combined. There are places where explanations could be given but are not, and one waits for the discussion but then they are not discussed and in general I would hope for greater insight into reaction conditions at the Nature Comm level. For example, is there a rationale for why NaOtBu was the optimal base? Why are sodium bases in general better than Li or K bases? Leading from this, was the reaction attempted without Pd? I imagine this is unlikely to be an uncatalyzed reaction, but in the presence of four equivalents of base, the control experiment should be done. In addition, is there a reason four equivalents are required? I can see this is often the case for small scale reactions, but what happens upon scaling the reaction up?

The supporting information is detailed, and spectra are of very high quality.

Once the questions of scope, control experiments and some small re-writing are addressed, the paper would meet the standards for publication in Nature Communications.

Other small issues include:

Table 1 is very hard to follow and looks more appropriate for supporting information rather than the text of the paper. If the authors want to leave it in the paper, could they divide it in sections so it is

easier to read?

Table 2 has two "d" footnotes in the footer

Line 154: typo: " and 6-qunolinyl"

Response to Reviewer Comments

Reviewer #1 (Remarks to the Author):

The authors report palladium-catalyzed mono- or bis-arylation of aryl(4-pyridyl)methanes or diaryl(4-pyridyl)methanes affording tetraarylmethanes. Tetraarylmethanes are important synthetic building blocks in modern organic chemistry, and a number of approaches to this important class of compounds have been developed. Catalytic arylation of the sp³-hybridized CH bonds of diarylmethanes or triarylmethanes would be potentially the most straightforward process. In this manuscript, the authors achieved the cross-coupling reactions of aryl(4-pyridyl)methanes or diaryl(4-pyridyl)methanes with aryl chlorides under simple palladium/phosphine catalysis. The coupling reactions proceeded smoothly with a variety of aryl chlorides having an electron-donating or -withdrawing group. This is an interesting piece of synthetic organic chemistry. However, I hesitate to recommend Publication in Nature Commun. because of the following reasons.

1) The nucleophilic coupling partner must have a 4-pyridyl moiety, which is a significant limitation that diminishes the value of this transformation.

Response: We thank the reviewer for their support and careful assessment of our manuscript. We agree with the reviewer that our method in the original submission was somewhat limited in that nucleophilic coupling partner had a 4-pyridyl moieties. Based on the reviewer's comments we investigated other nucleophilic partners such as 2-benzothiazolyl, 2-benzoxazolyl and 2-pyridyl. Substrates containing these group underwent coupling with aryl chlorides and then gave desired tetraarylmethane derivatives in good yields. We added these results to the manuscript (Table 5) and Supplementary Materials.

2) Although tetraarylmethanes are important in organic chemistry, the impact of these molecules is rather limited. It is necessary to add one or two examples that justify that this transformation is really useful.

Response: The comments of the reviewer inspired us to think more broadly about applications, and as a result we have modified the introduction. Tetraarylmethanes and related derivatives are important building blocks, with uses ranging from molecular devices to porous organic frameworks and applications from protein translocation detection to drug delivery (Amabilino, et al. *J. Chem. Soc., Chem. Commun.*, **1995**, 751; Ashton, et al. *Chem. Eur. J.*, **2000**, *6*, 3558; Dey, et al. *Chem. Sci.*, **2011**, *2*, 1046; Liao, et al. *Inorg. Chem.*, **2015**, *54*, 4029; Dong, et al. *Chem.*

Commun., **2014**, *50*, 14949; Lu, et al. *J. Mater. Chem. A.*, **2014**, *2*, 13831; Peschko, et al. *Chem. Eur. J.*, **2014**, *20*, 16273; Bonardi, et al. *Proc. Natl. Acad. Sci. U S A.*, **2011**, *108*, 7775; Huang, et al. *Langmuir.*, **2013**, *29*, 3223). Kato reported one liquid-crystalline (LC) molecule of bowl-shaped that form columnar and micellar cubic structure, using triary(4-pyridyl)methane moieties as building blocks. In order to apply our method, we synthesized the same compound from the reaction product **4q** (Table 3). It is noteworthy that Kato synthesized triary(4-pyridyl)methane was based on Friedel-Crafts arylations from diaryl(4-pyridyl)methane with 46% yield, which significantly less than our method (91% yield). We have added this result to the manuscript.

Additional minor comments:

1) At least, the outcomes of the coupling reactions with 2- and 3-pyridyl substituted diarylmethanes or triarylmethanes should be described. Discussion about the effect of the 4-pyridyl moiety is necessary.

Response: As discussed above, we investigated 2-benzothiazolyl, 2-benzoxazolyl and 2-pyridyl substituted triarylmethanes, which can be coupled with aryl chlorides to afford the tetraarylmethane derivatives under our optimized reaction conditions. These results indicate that the method is not limited to 4-pyridyl derivatives. When 2-benzyl pyridyl was used with PCy₃, triarylmethane formed in 82% yield and the tetraarylmethane only in 16% yield. Using cataCXium A, the triarylmethane yield was 73% and the tetraarylmethane formed in only 24% yield.

Unfortunately 3-benzyl pyridine did not undergo coupling under our current conditions (perhaps because of their high pKa) and will require the search for a new catalyst. Specifically, 3-benzyl pyridine was treated with the PCy₃-based catalyst, the triarylmethane was afforded only in 17% yield and 3-benzyl pyridine was recovered in 86% yield. No tetraarylmethane was observed.

When cataCXium A was used as ligand, we just recovered 3-benzyl pyridine in 99% yield. These results were not added to the manuscript.

2) How about aryl bromides, iodides, or sulfonates under the optimal reaction conditions?

Response: As mentioned in Table 1 (entry 2), 4-bromotoluene coupled with 4-benzylpyridine (77% assay yield). With cataCXium A as the ligand, 4-iodotoluene reacted with 4-benzylpyridine to afford tetraarylmethane product in 91% isolated yield. When p-tolyl trifluoromethanesulfonate was employed, both PCy₃ and cataCXium A based catalysts did not form triarylmethane or tetraarylmethane derivatives (mostly recovered starting materials).

We added the result with the aryl iodide to Table 1.

3) Page 2: 2aa and 2ba should be 2a and 2b.

Response: This typographical error has been corrected.

We thank the reviewer for their constructive feedback, and believe that the ensuing additions have strengthened the manuscript.

Reviewer #2 (Remarks to the Author):

In this manuscript, Zhang, Kim, Wu, Mao, and Walsh describe a powerful palladium-catalyzed deprotonative cross-coupling process to deliver tetraaryl alkanes. This reaction solves a current challenge in the preparation of tetraarylalkanes; it enables synthesis of these molecules with 4 different aryl substituents. It also does not require prefunctionalization of the benzylic

nucleophile; deprotonation generates the nucleophile in situ. The yields are excellent, as is the incorporation of heterocycles. The functional group tolerance is also fairly broad. The Supporting Information is thorough and well prepared. I anticipate that the synthetic organic and catalysis communities, as well as those using tetraarylalkanes in materials and biomedical applications, will be very excited by this new method. I recommend publication after the following very minor revisions have been addressed.

1) Given the discussion of solvent in the optimization, please give the solvent in Fig 1C for comparison.

Response: We thank the reviewer for their support of this work and on-point comments. We have since included a brief discussion of solvent in the optimization. We gave the solvent (xylene) in Fig 1C.

2) It is a distinct bond disconnection, but Crudden's method to make triaryl acetonitriles may be worth citing. It uses a transition metal-catalyzed approach to install the aryl substituents, and the nitrile can be converted to a nitrogen heterocycle, ultimately providing access to tetraarylmethanes with 4 distinct substituents.

Response: We agree that the Crudden/Nambo work is important. The paper was cited in the original submission (Ref.30)

3) In the first paragraph of the Results, it would be helpful to remind readers that the pyridyl group is necessary to attain the requisite acidity of the benzylic protons. Of course, it is also a desirable heteroaromatic from a synthetic perspective

Response: As we have added some other starting materials without pyridine moiety, we believe this point is better addressed in the revised manuscript.

4) In the Preliminary Catalyst Screen (line 77), the internal standard is CH_2Br_2 , but the Supporting Information says that 4,4'-di-tert-butylbiphenyl was used.

Response: We had omitted this fault, it should be 4,4'-di-tert-butylbiphenyl as the internal standard in preliminary catalyst screen. We revised it.

5) Table 2 has two footnote c's.

Response: This typo has been corrected.

6) How are triaryl starting materials (5) prepared? A brief comment would be helpful to show that these are readily accessible.

Response: Triaryl starting materials (5) prepared according to our group's previous method. We have added the following to the manuscript to clarify this to the readers:

(p. X, line XXX): "...Triarylmethanes are readily prepared by arylation of diphenylmethane derivatives using our prior approach..."

We thank the reviewer for their constructive feedback, and believe that the resulting additions have improved the manuscript.

Reviewer #3 (Remarks to the Author):

The manuscript "Palladium-Catalyzed Synthesis of Triaryl(4-pyridyl)methanes" by Walsh et al. describes the application of their deprotonative coupling methodology to the preparation of pyridyl-substituted triaryl methanes. The authors are certainly correct in their assertion that there are few if any metal catalyzed methods for the preparation of tetraaryl methanes. There is one that is missing from their list and that is the work of Nambo (ref 25) since high yielding routes to tetraaryl methanes are included in this publication, although limited to certain types of heterocycles as one of the rings that can be formed by a cycloaddition. The Walsh report is certainly the most general of the few methods available and one of the only general transition metal catalyzed routes.

The main criticism I have of the paper is the lack of any rationale for the inclusion of the pyridyl substituent. Is this necessary? There are no examples without pyridine in the starting material, leading to the question of whether this is needed for the reaction or not. In addition, the diversity of scope of the aryl groups incorporated is quite narrow, no furans or thiophenes are included and no pyrroles or N-heterocycles other than pyridine or indoles are included. In my opinion, these are the major negatives of the paper. The development of a route to tetraaryl methanes is highly impressive, but work should be done to really show the scope of the method since it is competing against existing methods with substrate limitations (organolithium and Friedel-Crafts).

Response: We thank the reviewer for their support of this work and thoughtful comments. In the original manuscript, there were no examples without pyridines in the starting materials, leading to the question that pyridine is needed for the reaction. Based on the comments, we screened some substrates without pyridines and found that 2-benzothiazolyl and 2-benzoxazolyl substituted triarylmethanes were promising precursors with our catalyst system. In the case of

broadening the aryl chlorides we found that 5-chlorobenzothiophene and 1-(4-chlorophenyl)-1H-pyrrole proved suitable, furnished products 8c and 8d in 61 and 63% yield, respectively (Table 5).

(2) On this same vein, did the authors try a sequential, one pot bis arylation of the substrate? This would dramatically improve the method. There is no doubt that routes to unsymmetrical tetraaryl

methanes are infinitely more valuable than routes to symmetrical versions. One could envision using Walsh chemistry to affect back to back additions of different aryl halides in one pot, this should be attempted and the results reported on.

Response: We tried one pot bisarylation of the substrate. As showed in the Fig, we chose 4-benzylpyridine as the starting material, and then added 4-chlorotoluene and 4-chlorobenzophenone without isolation of intermediates. The results were not promising. When the reactions conducted on a 0.2 mmol scale using Pd(OAc)₂ (5 mol %), cataCXium A (10 mol %), 1 (1 equiv), 2c (1 equiv) and NaOt-Bu (1 equiv) in 1,4-dioxane (0.2 M) at 100 °C, after 12 h, 2g (2 equiv) and NaOt-Bu (2 equiv) added to the reaction solution. After 12 h, worked up the reaction we found: 4b in 24% yield, 4g in 28% yield, 6e in 33% yield. If NaOt-Bu (3 equiv) were added at the reaction outset, the results were again poor: 4b in 34% yield, 4g in 25% yield, 6e in 23% yield. Thus, at this point this approach appears complicated and we opted to abandon it for now.

(3) In this reviewer's opinion, the paper would read more cleanly if results and discussion were combined. There are places where explanations could be given but are not, and one waits for the discussion but then they are not discussed and in general I would hope for greater insight into reaction conditions at the Nature Comm level. For example, is there a rationale for why NaOtBu was the optimal base? Why are sodium bases in general better than Li or K bases? Leading from this, was the reaction attempted without Pd? I imagine this is unlikely to be an uncatalyzed reaction, but in the presence of four equivalents of base, the control experiment should be done. In addition, is there a reason four equivalents are required? I can see this is often the case for small scale reactions, but what happens upon scaling the reaction up?

Response: We have combined the results and discussion. From the optimization process it appears that LiO-tBu and LiHMDS were not strong enough to deprotonate the benzylic protons thoroughly. These bases form stronger oligomers. When changed the bases to KOt-Bu, NaHMDS and KHMDS, we found the reaction products would decompose. It is not clear why.

As mentioned in the manuscript, in the absence of the palladium catalyst, 4-chlorobenzophenone and 4-benzylpyridine gave 33% AY of the triarylmethane and no tetraarylmethane product. Indicating that the Pd catalyst is needed to generate the product. We also tried to reduce the amounts of base, but we found that in some cases the triarylmethanes led to tetraarylmethanes in reduced yields. To demonstrate the potential utility of our method, we performed a gram scale reaction of 4-benzylpyridine 1a with 4-chlorobenzophenone 2g. The desired product 4g was obtained in 79% yield (1.88 g, Figure 4), demonstrating the reaction is scalable.

The supporting information is detailed, and spectra are of very high quality.

Once the questions of scope, control experiments and some small re-writing are addressed, the paper would meet the standards for publication in Nature Communications.

Other small issues include:

1) Table 1 is very hard to follow and looks more appropriate for supporting information rather than the text of the paper. If the authors want to leave it in the paper, could they divide it in sections so it is easier to read?

Response: Table 1 includes the optimized conditions and results, which are of interest to many synthetic chemists. Thus, we have followed the second option of the reviewer and we have divided it in two sections as Table 1 (mainly focused on catalysts loading, aryl halides and ligands screening) and Table 2 (focused on solvents, bases, ratio screening).

2) Table 2 has two "d" footnotes in the footer

Response: Corrected.

3) Line 154: typo: "and 6-quinolinyll"

Response: Corrected.

We thank the reviewer for their constructive feedback, and believe that the ensuing additions have strengthened our manuscript.

REVIEWERS' COMMENTS:

Reviewer #3 (Remarks to the Author):

The addition of other heterocycles is significant and removes a serious limitation of the initial submission. This, along with the other changes would make the work acceptable for publication with a few additional changes.

The addition of a comment on the Crudden-Nambo route to heterocyclic tetraaryl methanes is good, but considering that two of the three reviewers asked for this, and that the authors note it was "significant" it should be included in the Scheme in the opinion of this reviewer.

There seem to be a rather large number of typos included along with the changes. Those noticed by this reviewer are shown here, but the authors are encouraged to do a thorough review.

Reference 10 – good addition, has a small typo, ? and . back to back

Line 94: 4-iodotulene should be 4-iodotoluene

For Tables 1 and 2, CH₂Br₂ is still listed as internal standard, is this correct?

Line 114 entries¹ should have a space between entries and 1

Line 160, "Triarylmethanes are readily prepare" Should be "Triarylmethanes are readily prepared"

Response to Reviewer Comments

Reviewer #3 (Remarks to the Author):

The addition of other heterocycles is significant and removes a serious limitation of the initial submission. This, along with the other changes would make the work acceptable for publication with a few additional changes.

1) The addition of a comment on the Crudden-Nambo route to heterocyclic tetraaryl methanes is good, but considering that two of the three reviewers asked for this, and that the authors note it was “significant” it should be included in the Scheme in the opinion of this reviewer.

Response: We thank the reviewer for their support and careful assessment of our manuscript. We added the Nambo-Crudden chemistry to the Scheme (Figure 1F) in the manuscript.

2) There seem to be a rather large number of typos included along with the changes. Those noticed by this reviewer are shown here, but the authors are encouraged to do a thorough review. Reference 10 – good addition, has a small typo, ? and . back to back

Response: This typographical error has been corrected.

3) Line 94: 4-iodotulene should be 4-iodotoluene

Response: This typographical error has been corrected.

4) For Tables 1 and 2, CH₂Br₂ is still listed as internal standard, is this correct?

Response: As mentioned in the section-Preliminary catalyst screening results-we chose 4,4'-di-tert-butylbiphenyl as the internal standard for the UPLC analysis. For Tables 1 and 2, we chose CH₂Br₂ as the internal standard, because ¹H NMR was used to judge the assay yields. CH₂Br₂ is a singlet in the ¹H NMR and will not have overlap with the product peaks. Thus, CH₂Br₂ in the revised manuscript is correct.

5) Line 114 entries¹ should have a space between entries and 1

Response: This typographical error has been corrected.

6) Line 160, “Triarylmethanes are readily prepare” Should be “Triarylmethanes are readily prepared”

Response: This typographical error has been corrected.

We have carefully read the manuscript in search of other typos and corrected them. We thank the reviewer for her/his constructive feedback, and believe that the ensuing additions have strengthened the manuscript.